# Biotransformation of Platycosides, Saponins from Balloon Flower Root, into Bioactive Deglycosylated Platycosides

**DOI:** 10.3390/antiox12020327

**Published:** 2023-01-31

**Authors:** Kyung-Chul Shin, Deok-Kun Oh

**Affiliations:** 1Department of Integrative Bioscience and Biotechnology, Konkuk University, Seoul 05029, Republic of Korea; 2Department of Bioscience and Biotechnology, Konkuk University, Seoul 05029, Republic of Korea

**Keywords:** balloon flower root, platycosides, glycosidases, biological activity, deglycosylation methods, biotransformation

## Abstract

Platycosides, saponins from balloon flower root (Platycodi radix), have diverse health benefits, such as antioxidant, anti-inflammatory, anti-tussive, anti-cancer, anti-obesity, anti-diabetes, and whitening activities. Deglycosylated platycosides, which show greater biological effects than glycosylated platycosides, are produced by the hydrolysis of glycoside moieties in glycosylated platycosides. In this review, platycosides are classified according to the chemical structures of the aglycone sapogenins and also divided into natural platycosides, including major, minor, and rare platycosides, depending on the content in Platycodi radix extract and biotransformed platycosides. The biological activities of platycosides are summarized and methods for deglycosylation of saponins, including physical, chemical, and biological methods, are introduced. The biotransformation of glycosylated platycosides into deglycosylated platycosides was described based on the hydrolytic pathways of glycosides, substrate specificity of glycosidases, and specific productivities of deglycosylated platycosides. Methods for producing diverse and/or new deglycosylated platycosides are also proposed.

## 1. Introduction

*Platycodon grandiflorum* (Jacq.) A.DC., a perennial herbaceous flowering plant species [1], belongs to the family Campanulaceae, is commonly known as the balloon flower, and is used as a traditional herbal medicine for the treatment of cough, phlegm, sore throat, lung abscess, chest pain, dysuria, and dysentery [2]. In Northeast Asia, *P. grandiflorum* root (Platycodi radix), an edible vegetable, is widely used as a food supplement to make side dishes (seasoned balloon flower root), desserts (balloon flower root sweet), teas (balloon flower root tea), flavored liquor, and traditional herbal medicines for the treatment of pulmonary diseases and respiratory disorders, including sore throat, bronchitis, tonsillitis, asthma, and tuberculosis [2,3,4,5].

Platycosides, saponins of Platycodi radix, are the main components of Platycodi radix extract and the principal components responsible for these biological activities in *P. grandiflorum* [2,3]. They possess a wide range of biological properties, including not only antioxidant [6,7,8,9] but also anti-inflammatory [9,10,11,12,13], anti-tussive [14,15], anti-proliferative [16,17,18,19,20,21,22,23,24,25], anti-obesity [26,27,28,29,30,31,32], anti-diabetes [33,34,35], anti-allergic [36,37], immunomodulatory [38,39,40,41], cardiovascular protective [42,43,44], hepatoprotective [45,46], and whitening [9,47] effects. Platycosides are glycosylated compounds consisting of a non-glycoside component (aglycone) with a pentacyclic triterpene skeleton, and C-3 and C-28 glycoside groups (glycone) with 4~7 molecules, such as β-d-glucopyranose, β-d-apiofuranose, β-d-xylopyranose, α-l-rhamnopyranose, and α-l-arabinopyranose (Figure 1).

The absorption of glycosylated saponins in the gastrointestinal tract is poor. In contrast, deglycosylated saponins are more readily absorbed in the human gastrointestinal tract into the bloodstream and function as active compounds because of their lower molecular weight and greater hydrophobicity [48,49]. The biological activities of deglycosylated saponins are superior to those of glycosylated saponins [4,48]. Therefore, many studies have focused on the hydrolysis of the glycoside moieties in glycosylated saponins using chemical, physical, and biological transformation techniques [48,50]. Platycosides from *P. grandiflorum* detected by high-performance liquid chromatography (HPLC), liquid chromatography-mass spectrometry (LC-MS), and nuclear magnetic resonance (NMR) contain more than 4 glycosides; thus, platycosides with less than 4 glycosides are few or absent in Platycodi radix [4,8,51,52,53,54,55,56,57], indicating that natural major platycosides are highly glycosylated saponins compared to natural ginseng saponins with 2~4 glycosides [48,49,58]. To acquire bioactive deglycosylated platycosides, hydrolysis of the glycoside moieties in natural glycosylated platycosides is required.

Reviews on platycosides have mainly focused on their pharmacological effects [2,3,53,59]. However, reviews specifically focusing on the biotransformation of platycosides have not been reported. The current review describes the transformation methods of platycosides by hydrolysis of their glycoside moieties for the production of bioactive deglycosylated platycosides. Furthermore, we propose the production of diverse and/or new deglycosylated platycosides using glycosidases.

## 2. Classification of Platycosides

Platycosides are structurally classified into platycodigenin (PDN)-, polygalacic acid (PGA)-, platyconic acid (PCA)-, methyl-PCA-, PCA lactone-, and other-type platycosides according to the chemical groups at C-4 of the aglycone sapogenins [2,3]. Platycosides are also divided into natural platycosides, including major, minor, and rare platycosides (depending on their content in Platycodi radix extract), and biotransformed platycosides. Platycosides determined as distinct peaks by HPLC-UV can be considered major platycosides, while platycosides detected by only HPLC-evaporative light scattering detector (ELSD) or LC-MS except for platycosides detected by HPLC-UV can be considered minor platycosides. In addition, residual platycosides as new unusual platycosides in nature can be defined as rare platycosides.

### 2.1. Chemical Structural Classification

Platycosides are pentacyclic triterpene sapogenins containing two glycosidic side chains at C-3 and C-28 (Figure 1). The aglycone sapogenins of platycosides contain methyl, carboxyl, hydroxymethyl, and other groups of C-4 (R_1_). In nature, one side chain is a moiety of 1~3 glucoside molecules (β-d-glucopyranosyl-β-d-glucopyranosyl-β-d-glucopyranosyl residues) linked to C-3 (R_2_) of the sapogenin by a glycosidic bond, whereas the other chain is an oligosaccharide moiety of arabinoside, rhamnoside or acetyl-rhamnoside, xyloside, and apioside sequentially (β-d-apiofuranosyl-β-d-xylopyranosyl-α-l-rhamnopyranosyl or acetyl-rhamnopyranosyl-α-l-arabinopyranosyl residues) bound to C-28 (R_3_) in the aglycone by an ester linkage [4]. Thus, platycosides have diverse chemical structures according to the groups of C-4 and the types and numbers of the glycosides attached to C-3 and C-28.

Depending on the chemical groups at C-4 of sapogenins, platycosides are divided into five types, including PDN- (R_1_ = CH_2_OH), PGA- (R_1_ = CH_3_), PCA- (R_1_ = COOH), methyl-PCA- (R_1_ = COOCH_3_), and PCA lactone- (R_1_ = -COO-, linked to C-2), and the other-type platycosides with different groups at C-2, C-14, C-16, C-17, and C-21 [2,3]. Although the contents of specific platycosides in Platycodi radix extracts depend on *P. grandiflorum* species, cultivation regions, and solvent extraction methods, the contents of PDN- (compounds 1–30), PGA- (compounds 31–56), PCA- (compounds 57–67), and other-type platycosides to total platycosides are 50−55%, 30−35%, 7−13%, and >1%, respectively [4,8,51,52,55]. PDN-, PGA-, and PCA-type platycosides are major and minor platycosides, whereas methyl- PCA-, PCA lactone-, and the other-type platycosides are rare platycosides.

### 2.2. Natural Platycosides

There are 11 major platycosides as determined by HPLC-UV, LC-MS, or NMR: platycoside E (PE), platycodin D_3_ (PD_3_), platycodin D (PD), deapiosylated PE (deapi-PE), deapi-PD_3_, deapi-PD, platycodin A (PA), polygalacin D_3_ (PGD_3_), polygalacin D (PGD), 3”-*O*-acetyl polygalacin D_3_ (AcPGD_3_), and platyconic acid A (PCAA) [54,60,61,62,63,64]. These major platycosides have been used as substrates for the production of deglycosylated platycosides by glycosidases. There are 11 minor platycosides as detected by only HPLC-ELSD or LC-MS: 2″-*O*-acetyl platycodin D_2_, 3″-*O*-acetyl platycodin D_2_, platycodin D_2_, deapi-platycodin D_2_, 2″-*O*-acetyl platycodin D, 3″-*O*-acetyl platycodin D, polygalacin D_2_, 2″-*O*-acetyl polygalacin D_2_, 2″-*O*-acetyl polygalacin D, 3″-*O*-acetyl polygalacin D (AcPGD), and 3″-*O*-acetyl platyconic acid A [55,57,65]. Based on the literature, 76 compounds of platycosides have been isolated from *P. grandiflorum* [3]. Thus, there are 54 reported compounds of rare platycosides.

### 2.3. Biotransformed Platycosides

Biotransformed platycosides are produced from natural major platycosides by the hydrolysis of the glycoside moieties using glycosidases, including recombinant enzymes, commercial enzymes, purified enzymes provided by reagent companies, crude enzymes, and cells containing enzymes with reactions such as partial deglucosylation of only outer and middle glucosides [51,62,66,67,68,69,70], deglucosylation [4,71], deglucose-apiose-xylosylation [72], deglucose-apiose-xylose-rhamnosylation [73], deapiosylation with partial deglucosylation [8], deapiose-xylosylation with partial deglucosylation [74,75,76], and deapiose-xylose-rhamnose-arabinosylation with partial deglucosylation [52] (Table 1). Nineteen biotransformed platycosides, including deglucosylated PD (deglc-PD), deglc-PA, deglc-AcPGD, deglc-PCAA, deapi-PGD, deapi-PA, deapi-PGD_3_, deapi-PGD, deapi-PCA, deapiose-xylosylated PD (deapi-xyl-PD), deapi-xyl-PGD, deglc-api-xyl-PD, deglc-api-PGD, deglc-api-xyl-PGD, glucosyl-PDN (glc-PDN), glc-PGA, glc-PCA, arabinosyl-PDN (ara-PDN), and ara-PGA, have been reported. In total, 18 biotransformed platycosides, including deapi-AcPGD_3_, deapi-AcPGD, deapi-xyl-PA, deapi-xyl-AcPGD, deapi-xyl-PCAA, deglc-api-PA, deglc-api-xyl-PA, deglc-api-AcPGD, deglc-api-xyl-AcPGD, deglc-api-PCAA, deglc-api-xyl-PCAA, glc-ara-PDN, glc-ara-PGA, glc-ara-PCA, ara-PCA, PDN, PGA, and PCA, were suggested by the reactions of reported enzymes with unused substrates, combination of reported enzymes, and new application of known enzymes to platycoside hydrolysis (blue color in Figure 1, Figure 2, Figure 3 and Figure 4). On the other hand, platycosides that could not be suggested by the reported glycosidases are marked with red color in Figure 1, Figure 2, Figure 3 and Figure 4. Currently, 37 platycosides, including biotransformed and suggested biotransformed platycosides, have been introduced.

## 3. Biological Activities of Platycosides

### 3.1. Health Benefits

Platycosides have a wide range of pharmacological activities, including anti-inflammatory [9,10,11,12,13], apophlegmatic and anti-tussive [14,15], anti-cancer [16,17,18,19,20,21,22,23,24,25], anti-obesity [26,27,28,29,30,31,32], anti-diabetes [33,34,35], anti-allergic [36,37], immunomodulatory [38,39,40,41], cardiovascular protective [42,43,44], and hepatoprotective effects [45,46] and have been used as traditional herbal medicines in the pharmaceutical industry. Their antioxidant properties have also made them supplements in the food industry [6,7,8,9]. In addition, platycosides have been applied as ingredients in the cosmetic industry due to their anti-inflammatory, antioxidant, and whitening activities [9,47].

In vitro and in vivo results for the evaluation of biological activities of platycosides such as PD, PD_3_, PGD, deapi-PD, platycoside mixtures, and biotransformed platycosides are summarized in Table 2. The biological activities of the platycosides have been investigated using in vitro assays, in vitro cell lines, and in vivo animal models. For instance, the whitening effect is examined by tyrosinase inhibition using a tyrosinase inhibitor assay kit [9], the anticancer effect is studied by the inhibition of cancer cell proliferation using cancer cells [19,21,22], and the anti-inflammatory effect was evaluated by the reduction of pro-inflammatory cytokine levels using a lipopolysaccharide-induced mouse [13]. In vivo experiments were performed using natural platycosides, mainly PD and Platycodi radix extract as platycoside mixtures, whereas in vitro assay experiments were performed using biotransformed platycosides.

### 3.2. Biological Activities with Deglycosylation

In vitro anti-inflammatory activities of platycosides are evaluated using a lipoxygenase inhibitory assay [4,8,9]. The anti-inflammatory activities of platycodigenin-type platycosides followed the order glc-PDN (one glycoside) > deapi-PD (four glycosides) > deglc-PD (four glycosides) > PD (five glycosides) > PD_3_ (six glycosides) > PE (seven glycosides), indicating that the anti-inflammatory activity of platycodigenin-type platycosides increase as the number of glycosides decrease.

The antioxidant activities of platycosides have been estimated using a Trolox equivalent capacity assay or a total oxidant-scavenging capacity assay [6,8,9]. The antioxidant activities of platycodigenin-type platycosides using the Trolox equivalent capacity assay follow the order glc-PDN (one glycoside) > deapi-PD (four glycosides) > PD (five glycosides) > PD_3_ (six glycosides) > PE (seven glycosides). The antioxidant activity determined using the total oxidant scavenging capacity assay follows the order platycodigenin (no glycosides) > deapi-PE (six glycosides) > PD (five glycosides) > PE (seven glycosides). Thus, the antioxidant activities of platycodigenin-type platycosides are increased by decreasing the number of glycoside residues.

The tyrosinase inhibitor assay has been used to determine whitening activity [9]. The whitening activities as tyrosinase inhibitory activities of the platycodigenin-type platycosides follow the order glc-PDN (one glycosides) > PD (five glycosides) > PD_3_ (six glycosides) > PE (seven glycosides) [9]. The whitening activities of platycosides improve as the number of glycosides attached to the aglycone sapogenin decreases. The anti-inflammatory, antioxidant, and whitening activities of platycodigenin-type platycosides determined using assay kits increase as the number of glycosides attached to sapogenin decrease.

## 4. Deglycosylation Methods for Saponins

### 4.1. Physical Methods

Deglycosylated ginseng saponins have been prepared by steam heating, microwave heating, sulfur fumigation, and high hydrostatic pressure [48,50,71,75]. The deglycosylation of saponins using steam heating increases with increasing treatment temperature and time. However, the amount and production rate of deglycosylated saponins by steam heating are small and slow, respectively. Microwave heating is used as an efficient time-saving method to show a higher deglycosylation rate than steam heating [77]. Sulfur fumigation is a cheap method to decrease the drying time and generate new saponins but is environmentally harmful [78]. These physical methods cause structural changes in sapogenin structures by hydrolysis, dehydration, and decarboxylation, producing byproducts or new-type saponins [79,80] and they show low selectivity and have yet to be used in the deglycosylation of platycosides. However, the physical method using high hydrostatic pressure is used together with biological method such as enzymatic conversion to increase the productivity of deglycosylated saponins by increasing the activity and stability of glycosidases [71,75].

### 4.2. Chemical Methods

Under acidic and high-temperature conditions, glycosylated saponins such as natural ginsenosides are converted into deglycosylated saponins. Deglycosylation by acid hydrolysis is dependent on the pH, acid concentration, acid type, and treatment temperature and time. Deglycosylation increased with an increase in treatment temperature and time; however, the optimal pH is different for each saponin type [50]. Acids used in chemical transformation of saponins include acetic acid, formic acid, citric acid, lactic acid, tartaric acid, and hydrochloric acid. Acid hydrolysis causes epimerization, dehydration, and hydration reactions, resulting in the production of deglycosylated saponins and new saponins [81].

Deglycosylation via alkaline hydrolysis increases with increasing pH, pressure, temperature, and time. Alkaline hydrolysis requires strict treatment conditions because strong treatment conditions can cleave sapogenins [50]. Alkaline hydrolysis is an efficient method for the production of aglycone ginsenosides, such as protopanaxadiol and protopanaxatriol, because the production of sapogenins from natural ginseng saponins by alkaline hydrolysis has resulted in an 80% yield [82]. Therefore, alkaline hydrolysis can be applied to the production of aglycone platycosides such as PDN, PGA, and PCA. Alkaline hydrolysis also involves side reactions, such as epimerization, cyclization, and hydroxylation. The disadvantages of chemical methods are byproduct formation and the generation of environmental pollution.

### 4.3. Biological Methods

Physical and chemical methods for producing deglycosylated saponins have several limitations, such as low selectivity, generation of by-products, non-environment-friendly processes, and high energy consumption. To overcome these disadvantages, biological methods including enzyme conversion, cell conversion, and fermentation have been proposed [49]. The enzymatic conversion of saponins has been performed using recombinant enzymes [4,83], commercial enzymes [52,84], reagent enzymes [62,66,70], and native crude enzymes [8,85]. The conversion using recombinant enzymes showed the highest production of saponins compared to other enzymatic conversions. However, the application of recombinant enzymes in the food industry is limited. Enzymatic conversion exhibits high specificity, yield, and productivity for saponin deglycosylation. Moreover, this method is increasingly being recognized as a useful tool for structural modifications. However, it is less economical than fermentation because it requires enzyme purification and results in the loss of enzymes from cells during purification [86].

Cell conversion is the production of deglycosylated saponins by reactions using grown or washed microbial cells mixed with saponins, whereas fermentation involves cultivation of growing cells fed with saponins. Intestinal bacteria, yeast, fungi, and soil microorganisms have been used for the transformation of saponins [8,48,50,67,73]. Microbial transformation by intestinal bacteria requires expensive medium and exhibits low yield and poor productivity. Microorganisms isolated from soil-cultivated saponin-containing plants exhibit higher yields and productivity than intestinal bacteria [87,88,89]. However, soil microorganisms can be applied to foods only if they have been designated as generally regarded as safe (GRAS). GRAS microorganisms typically include human intestinal bacteria, lactic acid bacteria, and fungi. GRAS fungi are more suitable for the production than lactic acid bacteria because they are easier to grow in cheaper media and exhibit higher yields and productivities. Fermentation is more cost-effective than cell or enzymatic conversion because it does not require enzyme purification steps and can use both intracellular and extracellular enzymes. However, the productivity of deglycosylated saponins by fermentation is lower than that by cell conversion or enzymatic conversion. To increase the productivity and avoid the inhibition of saponins to cells during fermentation, a new strategy, such as continuous feeding of saponins in a fermenter, is required.

## 5. Biotransformation of Platycosides

### 5.1. Hydrolytic Pathways for Specific Glycosides

The hydrolytic pathways of glycosylated platycosides by β-d-glucosidase, β-d-apiosidase, β-d-xylosidase, α-l-rhamnosidase, and α-l-arabinosidase, which hydrolyze glucoside residues linked to C-3 and apioside, xyloside, rhamnoside, or acetyl-rhamnoside, and arabinoside residues linked to C-28 in the aglycone platycosides, respectively, are designated as A, B, C, D or D’, and E, respectively (Figure 2, Figure 3 and Figure 4). The hydrolytic pathways of glycosylated platycosides by β-d-glucosidases acting on the outer, middle, and inner glucoside residues at C-3 are designated as A1, A2, and A3, respectively.

Hydrolytic pathways of PDN-type platycosides by glycosidases have been proposed (Figure 2). In these pathways, PE and PA are converted into PDN by the reactions A1, A2, A3, B, C, D, and E, and A3, B, C, D’, and E, respectively, and 26 PDN-type platycosides are involved. The PGA-type platycoside, PGD_3_ or AcPGD_3_, is converted into PGA by the reactions of A2, A3, B, C, D or D’, and E, respectively, and 24 PGA-type platycosides were involved in the proposed hydrolytic pathways (Figure 3). Hydrolytic pathways of the PCA-type platycosides are proposed from PCCA to PCA via eight intermediate platycosides (Figure 4). In total, 60 platycosides are involved in the hydrolytic pathways from glycosylated platycosides to sapogenins.

The seven natural platycosides, platycodin D_2_ (compound **3**), 2″-*O*-acetyl platycodin D_2_ (compound **4**), 3″-*O*-acetyl platycodin D_2_ (compound **5**), 2″-*O*-acetyl platycodin D (compound **7**), polygalacin D_2_ (compound **32**), 2″-*O*-acetyl polygalacin D (compound **36**), and 3″-*O*-acetyl platyconic acid A (compound **58**), are not involved in the hydrolytic pathways (Figure 1). The diverse specific hydrolytic pathways of glycosylated platycosides, including deapi-PE, PE, PD, PA, PGD_3_, AcPGD_3_, and PCAA, into deglycosylated platycosides by specific reagent, commercial, recombinant, and crude glycosidases, and cells containing glycosidase are summarized in Table 1.

### 5.2. Substrate Specificity of Glycosidases

The biosynthesis of deglycosylated platycosides by glycosidases with substrate specificity is shown in Table 1. β-Glucosidases with narrow substrate specificity for only the outer glucoside residue at C-3 by the reaction of A1 have not yet been reported. β-Glucosidases from *Aspergillus usamii* [68], *Caldicellulosiruptor bescii* [51], *C. owensensis* [69], and *Cyberlindnera fabianii* [67], cellulase from *Trichoderma reesei* [70], laminarinase from *Trichoderma* sp. [62], and snailase from snails [66] hydrolyze the outer and middle glucoside residues linked to C-3 of sapogenin by the reactions A1 and A2, respectively, but not the inner glucoside. β-Glucosidase from *C. bescii* converts PE, deapi-PE, PGD_3_, and AcPGD_3_ into PD, deapi-PD, PGD, and AcPGD, respectively. β-Glucosidase from *Dictyoglomus turgidum* and cellulase from *A. niger* as the commercial enzyme Pluszyme 2000P catalyze the hydrolysis of all three glucoside residues linked to C-3 of sapogenin by the reactions A1, A2, and A3 [4,71]. β-Glucosidase from *D. turgidum* converts PE, deapi-PE PGD_3_, AcPGD_3_ PA, and PCAA into deglc-PD, deglc-api-PE, deglu-PGD, deglu-AcPGD, deglu-PA, and deglu-PCAA, respectively. The β-glucosidases used in the biotransformation of platycosides hydrolyze the glucoside moiety at C-3 of sapogenin but not the glycoside moiety at C-28.

Crude enzyme from *Rhizopus oryzae* hydrolyzes not only the outer and middle glucoside residues at C-3 by the reactions A1 and A2 with β-glucosidase activity, respectively, but also the outer apioside at C-28 by the reaction of B with β-d-apiosidase activity [8]. The crude enzyme converts PE, PGD_3_, PA, and PCAA into deapi-PD, deapi-PGD, deapi-PA, and deapi-PCAA, respectively. The crude enzyme can be applied to the conversion of the unused substrates AcPGD_3_ and AcPGD into deapi-AcPGD_3_ and deapi-AcPGD, respectively. Pectinase from *A. niger* as the commercial enzyme Cytolase PCL5 hydrolyzes the outer and middle glucoside residues at C-3 by the reactions A1 and A2 with β-glucosidase activity, respectively, and the apioside and xyloside residues at C-28 by the reactions B and C, with β-d-apiosidase and β-d-xylosidase activities, respectively [74]. The commercial enzyme converts PE and PGD_3_ into deapi-xyl-PD and deapi-xyl-PGD, respectively. The enzyme can also be applied to the conversion of unused substrates AcPGD_3_, PA, and PCAA into deapi-xyl-AcPGD, deapi-xyl-PA, and deapi-xyl-PCAA, respectively. α-l-Rhamnosidases, which hydrolyze rhamnoside in platycosides by the reaction of D, have not been used in the biotransformation of platycosides to date. α-l-Rhamnosidases hydrolyze rhamnosides, but not glucosides [35], suggesting that these enzymes can be applied to the production of new platycosides glc-ara-PDN, glc-ara-PGA, glc-ara-PCA, and ara-PCA from deapi-xyl-PD, deapi-xyl-PGD, deapi-xyl-PCAA, and deglc-api-xyl-PCAA, respectively. Pectinase from *A. aculeatus* as the commercial enzyme Pectinex Ultra SP-L with broad substrate specificity hydrolyzes the outer and middle glucoside residues at C-3 by the reactions A1 and A2, respectively, and the oligosaccharide moiety (apiosyl-xylosyl-rhamnosyl or acetyl-rhamnosyl-arabinosyl residues) at C-28 by the reactions B and E with β-d-apiosidase and α-l-arabinosidase activities, respectively [52]. The enzyme converts PE, PGD_3_, AcPGD_3_, PA, and PCAA into glc-PDN, glc-PGA, glc-PGA, glc-PDN, and glc-PCA, respectively.

Crude enzyme from *A. tubingensis* has β-glucosidase, β-d-apiosidase, and β-d-xylosidase activities [72]. The multi-hydrolytic activities, which may be because the crude enzyme contains several glycosidases [85]. The crude enzyme hydrolyzes all three glucoside residues at C-3 by the reactions A1, A2, and A3 with *β*-glucosidase activity and the apioside and xyloside residues at C-28 by the reactions B and C with β-d-apiosidase and β-d-xylosidase activities, respectively. The crude enzyme converts PE and PGD_3_ into deglc-api-xyl-PD and deglc-api-xyl-PGD, respectively. The crude enzyme can also be used to convert the unused substrates PA, AcPGD_3_, and PCAA into deglc-api-xyl-PA, deglc-api-xyl-AcPGD, and deglc-api-xyl-PCAA, respectively. The use of a combination of enzymes or cells with multiple enzymes is a good tool to produce new highly deglycosylated platycosides. If the combination of *A. aculeatus* pectinase and *D. turgidum* β-glucosidase are applied to the hydrolysis of the glycosylated platycosides PE, PA, PGD_3_, AcPGD_3_, and PCAA, all residues linked to C-3 and C-28 will be hydrolyzed into PDN, PDN, PGA, PGA, and PCA, respectively, by the reactions A1, A2, A3, B, C, D or D’, and E. The human intestinal bacterium *Bacteroides* converts PD into ara-PDN by the reactions A1, A2, A3, B, C, and D [73]. The cells also convert deglc-api-PD into deglc-api-PGD by dihydroxylation, which is converted into api-PGD.

### 5.3. Quantitative Biotransformation

A summary of the quantitative biotransformation of glycosylated platycosides, such as PE, PA, PGD_3_, and PCAA, into deglycosylated platycosides (such as PD, deglc-PD, deapi-PD, deapi-PA, deapi-PGD, deapi-PCAA, deapi-xyl-PD, deapi-xyl-PGD, deglc-api-xyl-PD, deglc-api-xyl-PGD, glc-PDN, glc-PGA, and glc-PGA) by glycosidases, including reagent, recombinant, commercial, and crude glycosidases, is shown in Table 3. Quantitative biotransformation is essential for industrial production of deglycosylated platycosides. In the production of PD from PE, snailase shows the highest concentration and volumetric productivity among the PD-producing *β*-glucosidases [66]. However, the specific productivity of PD by recombinant β-glucosidase from *C. bescii* is significantly higher than that of PD by snailase [51]. Recombinant β-glucosidase from *D. turgidum* shows the highest specific productivity for the production of deglycosylated platycosides among the deglycosylated platycoside-producing enzymes [4]. These results indicate that the recombinant glycosidases are the most efficient biocatalysts for platycoside hydrolysis compared to reagent, commercial, and crude glycosidases. However, recombinant glycosidases have limited food applications.

Commercial enzymes such as Pluszyme 2000P [71], Cytolase PCL5 [74], and Pectinex Ultra SP-L [52] exhibit higher specific productivity and lower cost than reagents such as cellulase [70], laminarinase [62], and snailase [66], indicating that commercial enzymes are more feasible for the industrial production of deglycosylated platycosides. The specific productivities of the commercial enzymes followed the order Pluszyme 2000P (deglucosylation) > Cytolase PCL5 (deapiose-xylosylation) > Pectinex Ultra SP-L (deapiose-xylose-rhamnosylation). These enzymes convert PE (seven glycosides) to deglc-PD (four glycosides), deapi-xyl-PD (three glycosides), and glc-PDN (one glycoside). Therefore, the specific productivity decreases with increasing degree of deglycosylation.

## 6. Conclusions and Future Perspectives

In this review, we describe the biotransformation of platycosides based on their hydrolytic pathways, glycosidases, and specific productivity of deglycosylated platycosides. However, the number of reports on the transformation of balloon flower root saponins are significantly fewer than those of ginseng saponins. Therefore, transformation methods for glycosylated ginsenosides can be applied for the transformation of glycosylated platycosides to produce diverse deglycosylated platycosides. Physical methods, including steam heating, microwave heating, and sulfur fumigation [48,50], and chemical methods, including acid and alkaline hydrolysis, which have been used in the hydrolysis of glycosylated ginsenosides, can be applied to the hydrolysis of glycosylated platycosides [50,81]. 

Currently, the pool of glycosidases that hydrolyze glycosylated platycosides is too small. Cloning of glycosidase genes and discovery of natural glycosidases are needed to expand the pool to produce diverse platycosides. To produce new platycosides by specific hydrolysis of the glycoside moieties in platycosides, different types of glycosidases with narrow substrate specificities are required. For instance, the discovery of β-glucosidases that hydrolyze only the outer glucoside residue at C-3, and β-d-apiosidase, β-d-xylosidase, α-l-rhamnosidase, and α-l-arabinosidase, which hydrolyze apioside, xyloside, rhamnoside, and arabinoside, respectively, but no glucosides are needed. To obtain highly deglycosylated platycosides, such as glc-ara-platycodigenin and ara-platycodigenin, glycosidases with broad substrate specificity that hydrolyze platycosides through different pathways are also required.

Microorganisms involved in saponin transformation include soil fungi and bacteria, intestinal bacteria, and GRAS fungi [48,50]. The use of GRAS microorganisms and enzymes originating from GRAS microorganisms for saponin transformation is an appropriate method for food applications. Although glycosidases from GRAS fungi such as *A. aculeatus* [52], *A. niger* [74,75,76], *A. tubingensis* [72], *A. usamii* [68], *R. oryzae* [8], and *T. reesei* [70] are used in the biotransformation of platycosides, only one human intestinal bacterium, *Bacteroides* [73] is applied in biotransformation. Thus, more diverse GRAS human intestinal bacteria, such as lactic acid bacteria, should be applied in the biotransformation of platycosides because they can metabolize orally ingested saponins and can be used safely in a variety of foods [48,90].

We propose methods for producing diverse and/or new deglycosylated platycosides that may contribute to the development of biotransformation processes to produce bioactive deglycosylated platycosides. The biological effects of platycosides are mainly attributed to natural platycosides. However, the effects of biotransformed platycosides using cell lines and animal models have not yet been investigated. Recently, biotransformed platycosides have increasingly been reported. Thus, biotransformed platycosides will allow for the initiation of pharmacological studies at the cell and animal levels, and platycosides with improved bioactivity are expected to be developed through pharmacological studies.

## Figures and Tables

**Figure 1 antioxidants-12-00327-f001:**
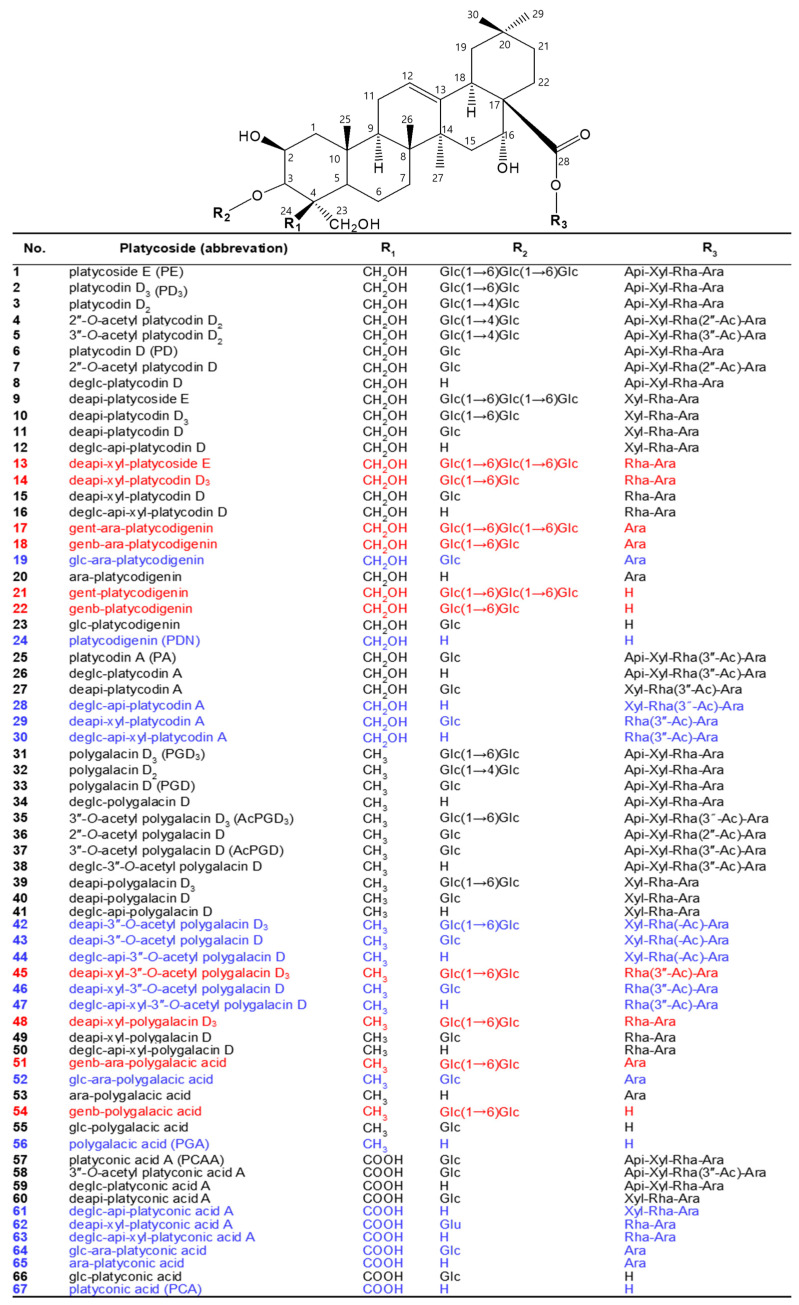
Chemical structures of major and minor platycosides and their glycoside moiety-hydrolyzed platycosides. The functional groups at C-4 (R_1_) of PDN-type (platycosides 1−30), PGA-type (31−56), and PCA-type (57−67) platycosides are CH_2_OH, CH_3_, and COOH, respectively. Platycosides contain glycosides at C-3 and C-28. The glycosides at C-3 (R_2_) are H, Glc, Glc-Glc, and Glc-Glc-Glc, whereas those at C-28 (R_3_) are H, Ara, Rha-Ara or Rha(-Ac)-Ara, Xyl-Rha-Ara or Xyl-Rha(-Ac)-Ara, and Api-Xyl-Rha-Ara or Api-Xyl-Rha(-Ac)-Ara. Glc, β-d-glucopyranosyl-; Api, β-d-apiofuranosyl-; Xyl, β-d-xylopyranosyl-; Rha, α-l-rhamnopyranosyl-; Ara, α-l-arabinopyranosyl-; Ac, acetyl; genb, gentiobiosyl; gent, gentiotriosyl; deapi, deapiosylated; deglc, deglucosylated; deapi-xyl, deapiose-xylosylated; deglc-api, deglucose-apiosylated; and deglc-api-xyl, deglucose-apiose-xylosylated. Blue indicates unreported platycosides, which can be suggested by the reported glycosidases. Red indicates unreported platycosides, which cannot be suggested by the reported glycosidases.

**Figure 2 antioxidants-12-00327-f002:**
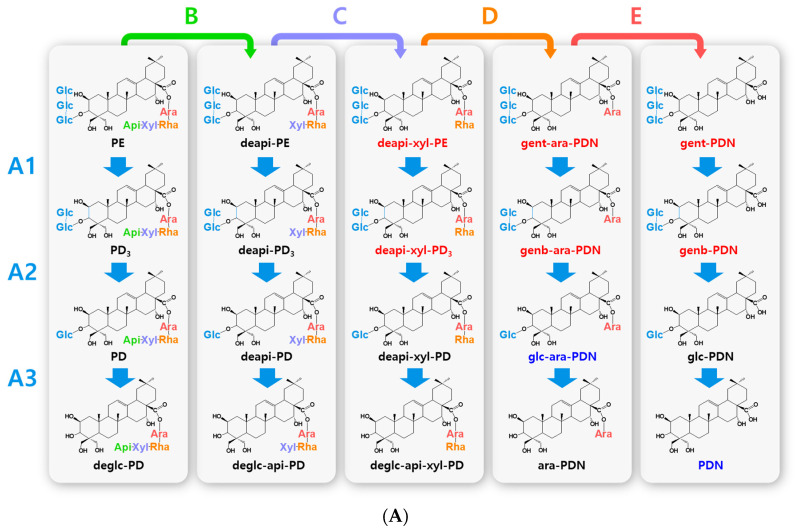
Hydrolytic pathways of the PDN-type platycosides by glycosidases. The hydrolytic pathways of the outer, middle, and inner glucoside residues linked to C-3 by β-d-glucosidases are assigned as A1, A2, and A3, respectively. The hydrolytic pathways of the four residues, including apioside, xyloside, rhamnoside or acetyl-rhamnoside, and arabinoside, linked to C-28 in PDN-type platycosides by β-d-apiosidase, β-d-xylosidase, α-l-rhamnosidase, and α-l-arabinosidase are assigned as B, C, D or D′, and E, respectively. (**A**) Hydrolytic pathways from PE to PDN. (**B**) Hydrolytic pathways from PA to PDN. R(-Ac), acetyl-α-l-rhamnopyranosyl-. Blue indicates unreported platycosides, which can be suggested by the reported glycosidases. Red indicates unreported platycosides, which cannot be suggested by the reported glycosidases.

**Figure 3 antioxidants-12-00327-f003:**
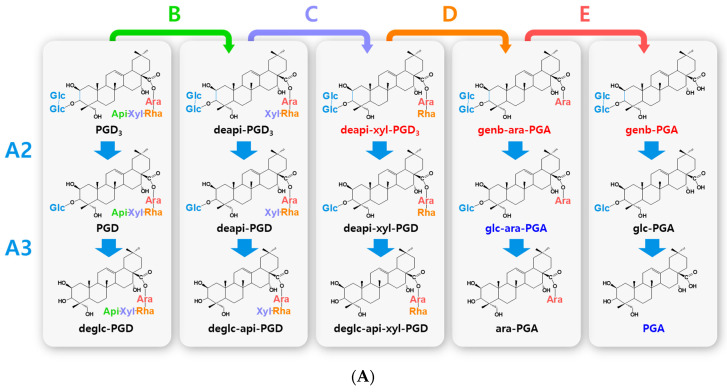
Hydrolytic pathways of the PGA-type platycosides by glycosidases. The hydrolytic pathways of the outer and inner glucoside residues linked to C-3 by β-d-glucosidases are assigned as A2 and A3, respectively. The hydrolytic pathways of the four residues, including apioside, xyloside, rhamnoside or acetyl-rhamnoside, and arabinoside, linked to C-28 in PDN-type platycosides by β-d-apiosidase, β-d-xylosidase, α-l-rhamnosidase, and α-l-arabinosidase are assigned as B, C, D or D′, and E, respectively. (**A**) Hydrolytic pathways from PGD_3_ to PGA. (**B**) Hydrolytic pathways from AcPGD_3_ to PGA. R(-Ac), acetyl-α-l-rhamnopyranosyl-. Blue indicates unreported platycosides, which can be suggested by the reported glycosidases. Red indicates unreported platycosides, which cannot be suggested by the reported glycosidases.

**Figure 4 antioxidants-12-00327-f004:**
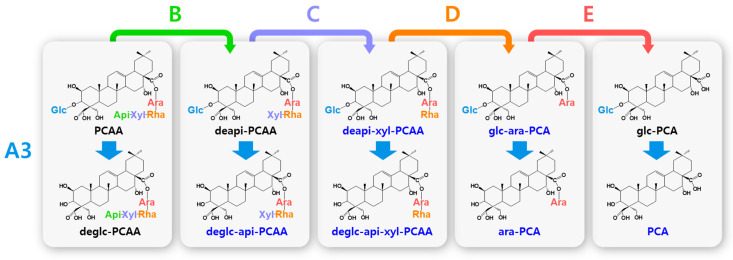
Hydrolytic pathways of the PCA-type platycosides by glycosidases. The hydrolytic pathways of the glucoside residue linked to C-3 by β-d-glucosidase are assigned as A3. The hydrolytic pathways of the four residues, including apioside, xyloside, rhamnoside, and arabinoside, linked to C-28 in PDN-type platycosides by β-d-apiosidase, β-d-xylosidase, α-l-rhamnosidase, and α-l-arabinosidase are assigned as B, C, D, and E, respectively. Blue indicates unreported platycosides, which can be suggested by the reported glycosidases. Red indicates unreported platycosides, which cannot be suggested by the reported glycosidases.

**Table 1 antioxidants-12-00327-t001:** Biosynthesis of biotransformed platycosides.

Biotransformed Platycosides	Reaction (Enzyme or Cells)	Pathway	Reference
Reported			
PD_3_, PD, deapi-PD_3_, deapi-PD, PGD, AcPGD	Partial deglucosylation of outer and middle glucosides (β-glucosidases, cellulase, laminarinase, snailase)	PE→PD_3_→PD, deapi-PE→deapi-PD_3_→deapi-PD, PGD_3_→PGD, AcPGD_3_→AcPGD	[51,62,66,67,68,69,70]
deglc-PD, deglc-PA, deapi-PGD, deglc-AcPGD, deglc-PCAA	Deglucosylation (β-glucosidases from *Dictyoglomus turgidum* and *Aspergillus niger*)	PE→PD_3_→PD→deglc-PD, deapi-PE→deapi-PD_3_→deglc-api-PD, PA→deglc-PA, PGD→deglc-PGD, AcPGD_3_→AcPGD→deglc-PGD, PCAA→deglc-PCAA	[4,71]
deapi-PE, deapi-PA, deapi-PGD_3_, deapi-PGD, deapi-PCA	Deapiosylation, partial deglucosylation (*Rhizopus oryzae* crude enzyme)	PE→PD_3_ or deapi-PE→PD or deapi-PD_3_→deapi-PD, PGD_3_→deapi-PGD_3_ or PGD→deapi-PGD, PCA→deapi-PCA	[8]
deapi-xyl-PD, deapi-xyl-PGD	Deapiose-xylosylation, partial deglucosylation (*Aspergillus niger* pectinase)	PE→PD_3_→PD→deapi-PD→deapi-xyl-PD, PGD_3_→PGD→deapi-PGD→deapi-xyl-PD	[74,75,76]
deglc-api-PD, deglc-api-xyl-PD, deglc-api-PGD, deglc-api-xyl-PGD	Deglucose-apiose-xylosylation (*Aspergillus tubingensis* crude enzyme)	PE→PD_3_→PD→deapi- or deglu-PD→deapi-xyl- or deglc-api-xyl-PD→deglc-api-xyl-PD, PGD_3_→PGD→deapi- or deglc-PGD→de-api-xyl- or deglc-api-PGD→deglc-api-xyl-PD	[72]
glc-PDN, glc-PGA, glc-PCA	Deapiose-xylose-rhamnose-arabinosylation, partial deglucosylation (*Aspergillus aculeatus* pectinase)	PE→PD→deapi-PD→glc-PDN, PA→deapi-PD→glc-PDN, PGD_3_ or AcPGD_3_→PGD or AcPGD→deapi-PGD or deapi-AcPGD→glc-PGA, PCAA→deapi-PCA→glc-PCA	[52]
ara-PDN, ara-PGA	deglucose-apiose-xylose-rhamnosylation (*Bacteroides*)	PD→deglc-PD→deglc-api-PD→ara-PDN, PD→deglc-PD→deglc-api-PD→deglc-api-PGD→ara-PGA	[73]
Proposed *^A^*			
deapi-AcPGD_3_, deapi-AcPGD	Deapiosylation, partial deglucosylation (*Rhizopus oryzae* crude enzyme)	AcPGD_3_→deapi-AcPGD_3_, AcPGD→deapi-AcPGD	
deapi-xyl-PA, deapi-xyl-AcPGD, deapi-xyl-PCAA	Deapiose-xylosylation, partial deglucosylation (*Aspergillus niger* pectinase)	PA→deapi-PA→deapi-xyl-PA, AcPGD_3_→deapi-AcPGD→deapi-xyl-AcPGD, PCAA→deapi-PCAA→deapi-xyl-PCAA	
deglc-api-PA, deglc-api-xyl-PA, deglc-api-APGD, deglc-api-xyl-AcPGD, deglc-api-PCAA, deglc-api-xyl-PCAA	Deglucose-apiose-xylosylation (*Aspergillus tubingensis* crude enzyme)	PA→deapi- or deglc-PA→deapi-xyl- or deglc-api-PA→deglc-api-xyl-PA, AcPGD_3_→deglu-AcPGD→deglc-api-AcPGD→deglc-api-xyl-AcPGD, PCAA→deapi- or deglu-PCAA→deapi-xyl- or deglc-api-PCAA→deglc-api-xyl-PCAA	
glc-ara-PDN, glc-ara-PGA, glc-ara-PCA, ara-PCA	Derhamnosylation (α-l-ramnosidase)	deapi-xyl-PD→glc-ara-PDN, deapi-xyl-PGD→glc-ara-PGD, deapi-xyl-PCAA→glc-ara-PCA, deglc-api-xyl-PCAA→ara-PCA	[35]
PDN, PGA, PCA	Deglycosylation (*Aspergillus aculeatus* pectinase + *Dictyoglomus turgidum* β-glucosidase)	PE→→glc-PDN→PDN, PA→→glc-PDN→PDN, PGD_3_ or AcPGD_3_→→glc-PGA→PGA, PCAA→→glc-PCA→PCA	

*^A^*, Platycosides that can be suggested by the reactions of reported enzymes with unused substrates, combination of reported enzymes, and new application of known enzymes to platycoside hydrolysis.

**Table 2 antioxidants-12-00327-t002:** Biological activity of platycosides.

Biological Activity	Compound (Assay, Cell Line, or Animal Model; In Vitro or In Vivo)	Key Findings	Reference
Antioxidant	Platycosides (total oxidant-scavenging capacity assay *^B^*; in vitro)	Peroxy radicals: PD (86) > PGA (60) > PDN (53) > deapi-PE (43) > PE (31) Peroxynitrite: PDN (235) > deapi-PE (127) > PD (102) > PE (75)	[6]
	Platycosides (total antioxidant capacity assay *^C^*; in vitro)	deapi-PD (0.145) > PD (0.092) > PE (0.037); deapi-PD (0.145) > deapi-PA (0.123) > deapi-PCA (0.106) > deapi-PGD (0.100); glc-PDN (0.68) > PD (0.15) > PD_3_ (0.07) > PE (0.05); glc-PDN (0.68) > glc-PCA (0.23) > glc-PGA (0.10)	[8,9]
Whitening	Platycosides (tyrosinase inhibitor assay *^D^*; in vitro)	glc-PDN (28) > PD (23) > PD_3_ (17) > PE (12); glc-PGA (39) > glc-PDN (28) > glc-PCA (7)	[9]
Anti-inflammatory	Platycosides (lipoxygenase inhibitory assay *^E^*; in vitro)	deglc-PD (55) > PD (47−49) > PD_3_ (44) > PE (39−41); glc-PDN (57) > deapi-PD (52) > PD (47−49) > PD_3_ (44) > PE (39−41); deapi-PA (59) > deapi-PGD (57) > deapi-PCA (53) > deapi-PD (52); glc-PGA (63) > glc-PCA (58) > glc-PDN (57)	[4,8,9]
	PGD (mouse macrophage cells; in vitro)	Inhibition of pro-inflammatory cytokine production	[11]
	PD (lipopolysaccharide- and cigarette smoke-induced mice; in vivo)	Reduction of pro-inflammatory cytokine levels, inhibition of lung inflammation	[12,13]
Anti-tussive	PD_3_, deapi-PD (sulfur dioxide-induced mice; in vivo)	Inhibition for the hypersecretion of airway mucin	[14]
	PD (ovalbumin-induced mice, in vivo)	Suppression of airway inflammation	[15]
Anti-proliferative	PD (human cancer cells, in vitro)	Inhibition for the proliferation of human cancer cells	[19,21,22]
	PD (xenograft mice, in vivo)	Suppression of tumor growth	[23,24,25]
Anti-obesity	PD (colorimetric assay, in vitro)	Inhibition of pancreatic lipase activity	[29]
	PD, platycosides *^A^* (diabetic and obese mice, in vivo)	Inhibition for the absorption of dietary fat, reduction of body weight	[27,28,30,31,32]
Anti-diabetic	Platycosides *^A^* (diabetic mice, in vivo)	Decrease of blood glucose levels, improvement of glucose homeostasis	[34,35]
Anti-allergic	Platycosides *^A^* (typical mice, in vivo)	Inhibition of allergic mediator production	[37]
Immunomodulatory	Platycosides *^A^* (immunosuppressed rats, in vivo)	Stimulation of immune responses and increase of immune cell production	[40]
	PD, PD_3_ (ovalbumin-induced mice, in vivo)	Enhancement of antigen-specific immune responses against viral infection	[39]
Cardiovascular protective	PD (hypertensive rats, in vivo)	Protection of cardiac tissue from hypertrophy and dysfunction	[42]
Hepatoprotective	PD (acetaminophen-induced mice, in vivo)	Protection of hepatocytes against hepatotoxicity	[45]

*^A^*, Platycodi radix extract; *^B^*, relative total oxidant-scavenging capacity value (%); *^C^*, trolox equivalent capacity value (mM); *^D^*, tyrosinase inhibition activity (%); *^E^*, lipoxygenase inhibition activity (%).

**Table 3 antioxidants-12-00327-t003:** Quantitative biotransformation of glycosylated platycosides into deglycosylated platycosides by glycosidases.

Enzyme	Substrate (mg/mL)	Product (mg/mL)	Molar Conversion Yield (%)	Volumetric Productivity (mg/L/h)	Specific Productivity (mg/g/h)	Reference
Reagent cellulase from *T. reesei*	*^E^* PE (1.0), PD_3_ (0.04)	PD (0.20)	100	0.85	NR	[65]
Reagent snailase from Snail digestive tract	*^E^* PE + deapi-PE	PD + deapi-PD (14.81)	100	673	4.5	[66]
β-Glucosidase from *C. fabianii*	*^E^* PE (1.0), PD_3_ (0.04)	PD (0.17)	43	2.42	NR	[67]
β-Glucosidase from *A. usamii*	*^E^* PE (1.0), PD_3_ (0.04)	PD (0.24)	99	122	41	[68]
Recombinant β-glucosidase from *C. bescii*	*^E^* PE (1.0), PD_3_ (0.04) PE (1.0)	PD (0.83) PD (0.79)	100 100	361 465	722 6640	[51]
Recombinant β-glycosidase from *C. owensensis*	*^E^* PE (5.0)	PD (3.95)	100	658	219	[69]
Recombinant β-glucosidase from *D. turgidum*	*^E^* PE (1.0), PD_3_ (0.04), PD (0.27) PE (1.0)	deglc-PD (0.96) deglc-PD (0.69)	100 100	48 99	9560 19,750	[4]
Reagent cellulase from *A. niger* (at 0.1 Mpa)	*^E^* PE (5.0), PD (0.07) PE (5.0)	deglc-PD (1.41) deglc-PD (0.41)	41 39	236 207	118 104	[71]
Reagent cellulase from *A. niger* (at 200 MPa)	*^E^* PE (5.0), PD (0.07) PE (5.0)	deglc-PD (3.49) deglc-PD (1.06)	100 100	582 532	388 354	
Crude enzyme from *R. oryzae*	*^E^* PE (3.5), PD_3_ (0.33), PD (1.27), deapi-PE (0.39), deapi-PD_3_ (0.13)	deapi-PD (3.89)	100	486	69	[8]
	*^E^* PA (0.96)	deapi-PA (0.76)	100	95	14	
	*^E^* PGD (2.22), PGD_3_ (0.97)	deapi-PGD (2.54)	100	318	45	
	*^E^* PCAA (0.85)	deapi-PCAA (0.69)	100	86	12	
Commercial pectinase from *A. niger*	PE (1.0) PGD_3_ (1.0)	deapi-xyl-PD (0.62) deapi-xyl-PGD (0.69)	100 100	52 46	103 92	[74]
Commercial pectinase from *A. niger* (at 150 Mpa)	PE (1.55)	deapi-xyl-PD (0.96)	100	240	481	[75]
Crude enzyme from *A. tubingensis*	PE (1.0) PGD_3_ (1.0)	deglc-api-xyl-PD (0.32) deglc-api-xyl-PGD (0.34)	62 60	32 43	64 85	[72]
Commercial pectinase from *A. aculeatus*	*^E^* PE (1.0), PD_3_ (0.04), PD (0.27), deapi-PE (0.07), deapi-PD (0.02)	glc-PDN (0.61)	70	17	1.7	[52]
	PE (1.0)	glc-PDN (0.42)	100	18	1.8	
	*^E^* PGD (0.8), AcPGD_3_ (0.16)	glc-PGA (0.21)	100	5.8	0.58	
	PGD_3_ (1.0)	glc-PGA (0.43)	100	18	1.8	
	*^E^* PCAA (0.17)	glc-PCA (0.10)	100	0.28	0.03	
	PCAA (1.0)	glc-PCA (0.55)	100	23	2.3	

*^E^*, extract containing platycosides; NR, not reported.

## Data Availability

Not applicable.

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
