# Peer review of "Biotransformation of Platycosides, Saponins from Balloon Flower Root, into Bioactive Deglycosylated Platycosides"

_antioxidants, 2023, doi:10.3390/antiox12020327_

Round 1
Reviewer 1 Report
This manuscript reviewed the structure and the bioactivities of platycosides from balloon flower root (Platycodi radix). The transformation methods for deglycosylation of saponins, including physical, chemical, and biological methods are also summarized. However, from my perspective, the manuscript must be improved before it be published.
1. Information of compounds 58-67 didn’t show in Figure 1.
2. Page 4, line 113: Section Natural platycosides, there are 11 major platycosides as determined by HPLC with a UV detector……
These compounds were determined by HPLC or NMR? I suspect that these compounds were determined by HPLC in reference 4, 8, 53, 54. 4, 8, 53, 54 reported the biotransformation of saponins, the 11 major and 11 minor platycosides are precursors or products? When discuss the types of structures, NMR is the best method to determine. I suggest the author cite the reference which reported the isolation and structure determination of platycosides.
3. Page 4, line 122 and 123: 76 types and 54 types, types or compounds?
4. Page 4, line 144: Currently, 113 platycosides, including natural, biotransformed, and suggested biotransformed platycosides, have been introduced.
This section described the Biotransformed platycosides, so the number of natural platycosides should be delete.
5. Page 7, line 214: The ‘.’ is missed at the end of the sentence.
6. Page 11, line 333: Caldicellulosiruptor should be abbreviated when it appear at the second time. Check all the other species.
7. Page 13, table 3: It is better if author give the yields.
Author Response
RE: antioxidants-2185032
Dear Editor
Thank you for forwarding us the comments of three reviewers. We thank all reviewers for their thoughtful comments. We have addressed all the questions and suggestions of the reviewers as described below.
Response to Reviewer #1
1) Information of compounds 58-67 didn’t show in Figure 1.
Response) Sorry for confusing due to our editorial issues. Figure 1 was corrected to include compounds 58-67 in the revised manuscript.
2) Page 4, line 113: Section Natural platycosides, there are 11 major platycosides as determined by HPLC with a UV detector……These compounds were determined by HPLC or NMR? I suspect that these compounds were determined by HPLC in reference 4, 8, 53, 54. 4, 8, 53, 54 reported the biotransformation of saponins, the 11 major and 11 minor platycosides are precursors or products? When discuss the types of structures, NMR is the best method to determine. I suggest the author cite the reference which reported the isolation and structure determination of platycosides.
Response) Thank you for your good comment. As you suggested, we added other references related to the determination of major platycosidesby LC-MS or NMR, and we modified the relevant sentence as follows: “There are 11 major platycosides as determined by HPLC-UV, LC-MS, or NMR : platycoside E (PE), platycodin D3 (PD3), platycodin D (PD), deapiosylated PE (deapi-PE), deapi-PD3, deapi-PD, platycodin A (PA), polygalacin D3 (PGD3), polygalacin D (PGD), 3²-O-acetyl polygalacin D3 (AcPGD3), and platyconic acid A (PCAA) [55,61-65].” (Line 114−117 of the revised manuscript)
3) Page 4, line 122 and 123: 76 types and 54 types, types or compounds?
Response) Thank you for your pointing out. To avoid misunderstandings due to incorrect wording, all "types" was changed to "compounds" in the revised manuscript. (Line 123−125 of the revised manuscript)
4) Page 4, line 144: Currently, 113 platycosides, including natural, biotransformed, and suggested biotransformed platycosides, have been introduced. This section described the Biotransformed platycosides, so the number of natural platycosides should be delete.
Response) Thank you for your suggestion. As you suggested, the number of natural platycosides was deleted and the relevant sentence was modified as follows: “Currently, 37 platycosides, including biotransformed and suggested biotransformed platycosides, have been introduced.” (Line 147−148 of the revised manuscript)
5) Page 7, line 214: The ‘.’ is missed at the end of the sentence.
Response) Thank you for your pointing out. We added the ‘.’ at the end of the sentence in the revised manuscript. (Line 221 of the revised manuscript)
6) Page 11, line 333: Caldicellulosiruptor should be abbreviated when it appear at the second time. Check all the other species.
Response) Thank you for your comment. As you commented, generic names of strains appearing at the second time in the manuscript were abbreviated. (Line 343, 349, 374, and 381 of the revised manuscript)
7) Page 13, table 3: It is better if author give the yields.
Response) Thank you for your suggestion. We have already indicated "yield" as "molar conversion" in Table 3. For better understanding, we changed "molar conversion" to "molar conversion yield" in Table 3 of the revised manuscript.
Response to Reviewer #2
1) Line 38: authors write "inflammatory". Is it anti-inflammatory? If yes… Please, correct.
Response) Sorry for the typo. We corrected it to "anti-inflammatory" in the revised manuscript. (Line 38 of the revised manuscript)
2) Lines 41-44: authors describe a generic platycoside molecule, without figure it is difficult to follow the text. Why don’t you insert a simplified figure of a generic platycoside?
Response) Thank you for your comment. These contents have already been shown in figure1. Therefore, we marked (Figure 1) at the end of this sentence, and the position of figure 1 was also adjusted. (Line 44 of the revised manuscript)
3) Lines 92-97: authors cited at least 67 compounds and three big groups. In figure 1 there are only 57. Please, insert the other compounds or correct the text or maybe write the reason of this difference.
Response) Sorry for confusing due to our editorial issues. Figure 1 was corrected to include compounds 58-67 in the revised manuscript.
4) Line 101 - Legend to the figure 1: a) please check the legend about the number of the compounds. b) probably, somewhere in the text its necessary a better explanation about blue and red platycosides.
Response) Thank you for your concern. a) As mentioned above, there was an editing problem with figure 1 and we corrected the figure 1. b) For better understanding, descriptions of blue and red platycosides were newly added to the text as follows: “(blue color in Figures 1–4). On the other hand, platycosides that could not be suggested by the reported glycosidases are marked with red color in Figures 1–4.” (Line 145−147 of the revised manuscript)
5) Table1 - Line 147: authors should better specify in the legend what is "proposed platycosides". Are them the "suggested" compounds cited at line 142? It is a bit confusing...
Response) Thank you for your comment. As you mentioned, "proposed platycosides" are the "suggested" compounds cited at line 142 in the original manuscript. Therefore, for better understanding, the footnote of Table 1 was newly added as follows: “A, Platycosides that can be suggested by the reactions of reported enzymes with unused substrates, combination of reported enzymes, and new application of known enzymes to platycoside hydrolysis.” (Line 151−153 of the revised manuscript)
6) Table 2 – line 172: Please, it is necessary to discuss some results reported in table 2. For example, the magnitude of the antioxidant capacity, anti-inflammatory activity, or anticancer activity... maybe there are other peculiar characteristics.
Response) Thank you for your comment. We have already explained the contents of anti-inflammatory, antioxidant, and whitening activities in “3.2. Biological activities with deglycosylation”. (Line 174−197 of the original manuscript or line 181−204 of the revised manuscript) We discussed these biological activities in relation to the degree of deglycosylation of platycosides, which is the subject of this paper, rather than other peculiar characteristics.
7) Line 174-197: In this paragraph the magnitude of the activity (difference is big or not?) can help to understand the point, e.g., better antioxidant activity (for example)
Response) Thank you for your good suggestion. For better understanding, instead of the paragraph you pointed out, we numerically indicated the magnitude of the activity in the column of key findings in Table 2, and added the related contents to the foot note as follows: “B, relative total oxidant-scavenging capacity value (%); C, trolox equivalent capacity value (mM); D, tyrosinase inhibition activity (%); E, lipoxygenase inhibition activity (%).” (Line 178−180 of the revised manuscript)
8) Line 322: Please, write "(compound 3)" instead of (3), same for others (4), (5), (7), (32), (36), (58).
Response) Thank you for your pointing out. As you mentioned, we added “compound” in front of those numbers in the revised manuscript. (Line 331−334 of the revised manuscript)
9) Line 335: Please write spp. if you mean more than 1 species.
Response) Thank you for your concern. What we mean is only one species.
10) Line 346: Please, write Rhizopus oryzae instead R. oryzae.
Response) Thank you for your pointing out. We corrected “R. oryzae” to “Rhizopus oryzae” in the revised manuscript. (Line 356 of the revised manuscript)
Response to Reviewer #3
1) In the Introduction, provide the full binominal name of the species with the name of its author - Platycodon grandiflorus (Jacq.) A.DC.
Response) Thank you for your comment. As you mentioned, we changed the name with the full binominal name of the species, Platycodon grandiflorus (Jacq.) A.DC. in the revised manuscript. (Line 26 of the revised manuscript)
2) Platycodi radix should be written in Italic (multiple times in the text of the manuscript);
Response) Thank you for your concern. “Platycodi radix” is not a scientific name, but a term referring to the root of balloon flower. Therefore, italic font is generally not used in all other papers.
3) To the HPLC method, please add information about the detector used (UV, DAD, or other) - line 53.
Response) Thank you for your comment. We have already written the detectors as “HPLC with a UV detector” and “HPLC with an ELSD”. (Line 72−74, 113, and 118−119 of the original manuscript) However, as you mentioned in the next question, we modified them to “HPLC-UV” and “HPLC-ELSD”. (Line 85−87, 114, and 119 of the revised manuscript)
4) Instead of LC/MS, I propose to use LC-MS and, similarly, for HPLC methods, HPLC-UV, HPLC-DAD, etc., respectively.
Response) Thank you for your comment. As you mentioned, we modified them to “LC-MS”, “HPLC-UV”, and “HPLC-ELSD”, respectively. (Line 66, 85−87, 114, 119, and 120 of the revised manuscript)
5) The term anti-cancer is better kept for cytostatics, for food-derived compounds it is better to use the term antiproliferative.
Response) Thank you for your suggestion. As you suggested, we changed the word “anti-cancer” to “anti-proliferative”. (Line 38 and Table 2 of the revised manuscript)
Finally, we would like to thank you and the reviewers again for thoughtful suggestions and comments for improvement of this manuscript, and we hope that the revised version of our manuscript meet the high standards of Antioxidants and is now acceptable for publication.
Thank you very much for your interest and assistance.
Sincerely,
Deok-Kun Oh, PhD
Department of Bioscience and Biotechnology, Konkuk University
120 Neungdong-ro, Gwangjin-gu, Seoul 05029, Korea.
Phone: 82-2-454-3118, Fax: 82-2-444-5518, Email: deokkun@konkuk.ac.kr
Reviewer 2 Report
This review entitled "Biotransformation of platycosides, saponins from balloon flower root, into bioactive deglycosylated platycosides” written by Kyung-Chul Shin and Deok-Kun Oh describes platycosides classification, biological activities of platycosides, methods for deglycosylation of saponins and biotransformation of glycosylated platycosides into deglycosylated platycosides. The review is clearly written, and all suggested arguments were described. The review is interesting but, it contains some flawn to be improved before acceptance. Points that need to be addressed.
Introduction
Line 38: authors write "inflammatory". Is it anti-inflammatory? If yes… Please, correct.
Lines 41-44: authors describe a generic platycoside molecule, without figure it is difficult to follow the text. Why don’t you insert a simplified figure of a generic platycoside?
Classification of platycosides
Lines 92-97: authors cited at least 67 compounds and three big groups. In figure 1 there are only 57. Please, insert the other compounds or correct the text or maybe write the reason of this difference.
Line 101 - Legend to the figure 1: a) please check the legend about the number of the compounds. b) probably, somewhere in the text its necessary a better explanation about blue and red platycosides.
Table1 - Line 147: authors should better specify in the legend what is "proposed platycosides". Are them the "suggested" compounds cited at line 142? It is a bit confusing...
Biological activities of platycosides
Table 2 – line 172: Please, it is necessary to discuss some results reported in table 2. For example, the magnitude of the antioxidant capacity, anti-inflammatory activity, or anticancer activity... maybe there are other peculiar characteristics.
Line 174-197: In this paragraph the magnitude of the activity (difference is big or not?) can help to understand the point, e.g., better antioxidant activity (for example)
Biotransformation of platycosides
Line 322: Please, write "(compound 3)" instead of (3), same for others (4), (5), (7), (32), (36), (58).
Line 335: Please write spp. if you mean more than 1 species.
Line 346: Please, write Rhizopus oryzae instead R. oryzae.
Author Response

(The authors gave the same response as above.)

Reviewer 3 Report
The topic of the reviewed manuscript seems to be interesting, however, in my opinion, the scope of the manuscript is slightly different from the topics of Antioxidants. This is a review paper based on broad and current scientific literature. The Authors have presented the gathered information for the proposed subject with great accuracy. I have only a few minor comments, which I put together below:
- In the Introduction, provide the full binominal name of the species with the name of its author - Platycodon grandiflorus (Jacq.) A.DC.
- Platycodi radix should be written in Italic (multiple times in the text of the manuscript);
- To the HPLC method, please add information about the detector used (UU, DAD, or other) - line 53;
- Instead of LC/MS, I propose to use LC-MS and, similarly, for HPLC methods, HPLC-UV, HPLC-DAD, etc., respectively;
- The term anti-cancer is better kept for cytostatics, for food-derived compounds it is better to use the term antiproliferative.
Author Response

(The authors gave the same response as above.)

Round 2
Reviewer 2 Report
The authors fully answered to the previous remarks, the manuscript can be published.